# Basic research investment, innovation capability improvement, and economic growth efficiency

Yichen Jiang[1], Bin Xu[1]*, Li Fang[1]*, Boyue Sun[2]*, Liyan Hu[3], Wang Hui[4]

1 Zhejiang Science and Technology Information Research Institute, Hangzhou, China, 2 School of Economics, Zhejiang Gongshang University, Hangzhou, China, 3 School of Economics, Shanghai University, Shanghai, China, 4 Taiwan Research Institute, Xiamen University, Xiamen, China

* 85283426@qq.com (BX); fangli1229@163.com (LF); 1332002630@qq.com (BS)

**Data Availability Statement:** All relevant data are within the manuscript and its Supporting Information files.

**Funding:** This paper was supported by the Zhejiang Science and Technology Information Research

## Abstract

This study explores the relationship between the proportion of basic research investment in R&D expenditure and regional economic growth efficiency and its influence path. A panel of data from 31 China's provincial was analysed by the fixed-effects model over 2009 to 2019. Empirical results highlight that: (1) the increase in the proportion of basic research investment has a significant promoting effect on regional economic growth efficiency, but it presents an "inverted U-shaped" impact path. Meanwhile, the threshold effect model test results reveal that basic research investment plays a more significant role in promotion when the per capita income level exceeds a certain threshold. (2) The test results of the mediating effect model indicate that the improvement of the level of human capital is the intermediate channel through which the investment in basic research affects regional economic growth efficiency, while the mediating effect of the increase of knowledge storage on the process of the investment in basic research affecting regional economic growth efficiency is not supported by data. Potential policy measures are discussed.

## 1. Introduction

Since its reform and subsequent opening up, China's economy has developed rapidly and has achieved the "China Miracle", which has attracted global attention. In this process, the "introduction, absorption, and re-digestion" model, represented by imported equipment, foreign direct investment (FDI) and purchase of foreign technology, has played an important role in improving China's technological level and ensuring rapid economic growth. However, as the gap between China's economic development and the world's frontier level continues to narrow, the late-developing advantage that could have been earned from the industrial and technological experience of developed countries has gradually disappeared. From the perspective of the external environment, developed economies strictly limit the spill-over of high-technology and advanced knowledge, retain high-end productive service industries in the process of international industrial division, outsource non-core businesses, and firmly grasp the key technologies in the industrial chain and the stagnant core technologies. This situation means that

Institute R&D Project: "A Data Processing Method for Intelligent Mining and Text Anomaly Monitoring"(R2021D001).

**Competing interests:** The authors have declared that no competing interests exist.

it is increasingly difficult to directly absorb the technology and knowledge of developed economies. From the perspective of the internal environment, China's economy has entered a new stage, and the previous simple technology import mode is beginning to appear 'powerless' for future industrial structure optimization. As the source of all scientific and technological innovation activities, basic research is not only the breakthrough point of various technical problems but also related to China's overall science and technology strategy. The 19th National Congress of the Communist Party of China pointed out that it is necessary to aim for the forefront of the world's scientific and technological development and to achieve breakthroughs in original achievements by strengthening basic research. As China's economy shifts to a high-quality development model, enhancing independent innovation capacity is undoubtedly seen as a core solution. Tightly linked to this view, increasing the proportion of basic research in scientific research has appeared as a promising pathway towards high-quality development model.

This paper aims at contributing to the literature through the following three aspects. First, from the research perspective, unlike the existing literature that focuses on the overall R&D activities or the total amount of basic research investment, this study attempts to explore the factors that may improve the efficiency of regional economic growth from the structural perspective of R&D resource investment. Second, in terms of theoretical analysis, this study constructs an analytical framework between the proportion of basic research investment in R&D activities and the efficiency of regional economic growth. Third, we use the panel data of China from 2009 to 2019 to empirical research. Because the technological level of China's economic development is getting closer and closer to the world's first-class level it is a suitable sample for this paper to discuss the economic effect of the increase in the proportion of basic research investment. Finally, in the empirical analysis, this study comprehensively uses the intermediary effect model, threshold effect model, and other empirical methods to carefully test the intermediary channels and various heterogeneous effects on the proportion of basic research investment that affects the efficiency of regional economic growth.

## 2. Literature review

As the source of all technological progress, the positive impact of R&D activities, especially basic research, has penetrated all aspects, levels, and fields of society, and has had a huge impact on the improvement of innovation ability, industrial structural adjustment, and regional economic growth. Closely related to the theme of this study is the economic effect of basic research investment. The existing literature on basic research primarily focuses on three aspects. First, several studies concentrate on the relationship between basic research investment and technological progress and innovation ability. Li examined the relationship between basic research investment and technological progress, innovation capacity, and labour rate based on the panel data of 30 countries in the world from 1998 to 2012 [1]. They concluded that basic research investment can effectively improve the TFP growth rate of the economy, and has a more significant promoting effect for countries with high income and high intellectual property protection. On this basis, Cai explored the impact mechanism of basic research activities on technological innovation in the field of power batteries [2]. Ye used the funding of the National Natural Science Foundation of China as an agent for basic research investment, and empirically tested the innovative transformation of basic research investment at the urban level [3]. Zeng ascertained the impact of basic research cooperation activities on enterprise innovation performance at the micro enterprise level, and found that the broader the scope of basic research cooperation, the lower the cooperation intensity, and the more conducive to the improvement of enterprise innovation performance [4]. Second, certain studies have

investigated the mechanism of basic research activities driving regional economic growth and efficiency improvement. Li and Zhu found that there is a significant R&D spill-over effect in the process of international trade under the open economic model, and further assessed the technical progress of industrial industries and the improvement of total factor productivity [5]. Zhang expanded Romer's innovation driven model by introducing two variables—basic research and applied research—and analysed the investment choices that local governments should make to promote economic growth through R&D activities [6]. Wangjuan and Ren xiaojing further explained the role of basic research activities in promoting industrial total factor productivity from the micro level [7]. Third, the latest literature began to pay attention to certain characteristics of basic research activities and appropriate institutional arrangements [8, 9]. For example, Sun and Ding (2022) focused on the layout and balance of basic research in China [10].

The extant literature comprises substantial theoretical research and empirical analyses concerning the efficiency of regional economic growth. Fu and Chen started from the perspective of technology introduction methods and used systematic GMM to empirically test the impact of imports, foreign direct investment, and the purchase of foreign technology authorization on China's economic growth efficiency from 2000 to 2013 [11]. Souare found that FDI significantly improved the level of total factor productivity by studying the impact of international industrial transfers on the technological innovation capability of the Canadian manufacturing industry [12]. Based on Mauritius' manufacturing development data. Fauzel demonstrated that international industrial transfers promoted the improvement of total factor productivity (TFP) of host countries through spill-over and cumulative effects [13]. In addition, some studies try to ascertain the efficiency of regional economic growth from the perspective of industrial intelligence [14, 15]. In recent years, scholars have noticed that garbage discharge, logistical efficiency, power consumption, and ICT industry development, among other factors are closely related to regional economic growth and industrial structure [16, 17]. In particular, the rapid development of ICT industry has increasingly become an important factor driving industrial restructuring and economic growth in recent years. Magazzino confirmed that the penetration of ICT technology is not only an important reason for the increase of electricity consumption and carbon dioxide emissions, but also promotes the local economic growth based on the panel data of 16 EU countries and a variety of econometric empirical methods [18]. For developing countries, ICT technology has also played a significant role in promoting economic growth. Latif focused on the dynamic relationship between ICT and economic growth in BRICS countries from 2000 to 2014, and found that there was a long-term elasticity between them, indicating that ICT development also drove high economic growth in developing countries [19]. In addition, scholars have also used numerical simulation techniques, such as machine learning algorithms and other tools, to study the causal relationship between the development of ICT industry on electric energy, urbanization development, pollution emissions and economic aggregate growth in OECD countries and other economies [20].

Although the literature has achieved fruitful results in basic research investment to promote economic growth, technological progress and improve regional innovation ability [21], most of these studies start from the perspective of the overall R&D resource investment or the total amount of basic research investment, and do not pay attention to the structural optimization of R&D investment. As R&D activity itself is a complex system that simultaneously includes basic research, applied research, and experimental development research, if the economic effect of R&D activity is investigated only from the total R&D investment itself, it is not only difficult to conduct more in-depth mechanism research and analysis on how R&D activities drive the improvement of economic efficiency but also easy to lead to the internal resource mismatch of R&D activities. Moreover, China is a typical developing country. In the past four

decades of rapid development, China's scientific research resources have been mainly used for applied research, while less resources have been allocated to basic research activities. However, with the total amount and quality of China's economic development approaching the world's cutting-edge level, the adjustment of industrial structure and the improvement of innovation ability will rely more on basic research activities. Therefore, adjusting the allocation structure of scientific research resources is very important and urgent for China's economic development and even the healthy development of the world economy. In the short term, the increase of the proportion of basic research means the decrease of applied research investment. What is the comprehensive impact on the efficiency of economic growth? Is there an inverted U-shaped relationship? The answers to these questions will not only help clarify the mechanism of the impact of basic research investment on economic growth efficiency in theory but also provide scientific guidance for regions to adjust the structure of R & D investment, optimize the allocation of scientific research resources, and boost high-quality economic development.

## 3. Theoretical mechanisms and research assumptions

From the perspective of resource scarcity theory, the total amount of resources invested in R&D research in a certain period of time is limited, which results in a competitive relationship between basic and applied research in resource allocation. Both basic and applied research activities contribute to improving economic growth efficiency. The former strengthens the absorption capacity of external technology [22], creates general knowledge, and accelerates knowledge spill-over [23]. Other pathways are involved in the process of technological progress. The latter improves the efficiency of regional economic growth by transforming basic research results, thus accelerating the application of disruptive technologies and transforming advanced foreign technologies [24, 25].

Therefore, the impact of the increase in the proportion of basic research investment on the efficiency of economic growth is likely to promote and inhibit effect simultaneously. It is mainly because, on the one hand, if the investment in basic research increases rapidly in the short term, it will crowd out the investment in applied research resources, which will be detrimental to the development of applied research. Gersbach and other scholars have also found in their research on the R&D spill-over effect, with the help of the expansion model of innovative product categories and the hierarchical model including basic research and applied research, that the rapid rise of basic research investment will crowd out the resource investment in applied research and reduce the balanced growth rate [26]. Especially, basic research itself has the characteristics of long cycle, strong uncertainty, and large investment, and its role in promoting economic growth needs a long transformation cycle. However, the applied research and experimental development activities have a direct promoting effect on the transformation of scientific and technological achievements and the improvement of enterprise benefits in the short term. Therefore, when limited R&D resources are allocated more to the basic research link, it is bound to crowd out investment in applied research, which will have a negative impact on the efficiency of economic growth. However, on the other hand, the development of basic research activities has a direct and positive role in promoting human capital, enhancing innovation ability, and expanding knowledge stock. These factors are key variables for endogenous economic growth and productivity improvement [27, 28]. Basic research is the source of all science and technology, and plays an irreplaceable role in strengthening and supplementing the R&D innovation industry chain. It can not only create new research content and research directions for applied research but also overcome theoretical bottlenecks and technical obstacles encountered in applied research, thereby indirectly improving the efficiency of regional economic growth. Zhang confirmed that basic research can not only directly promote economic growth but also indirectly promote the breakthrough and development of

applied research in improving the economic growth rate through vector auto-regression, the Granger causality test, and other methods [29]. Without a breakthrough in basic research, there will be no way to start the application and experimental development in the research content, and there will be a lack of advanced theoretical guidance in the research methods. Moreover, due to its non-utilitarian characteristics, basic research activities can provide a positive external environment for enterprise R&D innovation and process progress and upgrading through information and knowledge spill-over effects.

Considering that the proportion of applied research in China's technology catch-up strategy over the years is much higher than that of basic research, many scholars also pointed out that China's current economic development process has reached the stage where we must pay attention to the input of basic research in the process of economic growth [30]. Therefore, an appropriate increase in investment in basic research can optimise the distribution of resources in various departments within the R&D system, subsequently promote the efficiency of "basic research—applied research—industrial application", and finally improve the overall efficiency of industrial core technology innovation. However, when the proportion of basic research investment is excessively increased, a resource mismatch can easily occur within the R&D system, thus reducing the overall technology development and industrial application iteration speed, which is not conducive to the improvement of regional economic growth efficiency.

**Hypothesis I:** In terms of both promotion and inhibitory effects, the impact of basic research investment on the efficiency of economic growth may follow an inverted U-shaped path.

Schumpeter's endogenous growth theory emphasises that the creative destruction mechanism, triggered by technological innovation, is one of the core driving forces for industrial restructuring and endogenous economic growth. Basic research can improve the innovation ability of economic subjects by cultivating human capital and enriching basic knowledge, and then producing the "Schumpeter effect", which is manifested in technological progress and advanced adjustment of industrial structure. It is conducive to enhance the economic growth effect through the improvement of production efficiency and intensive utilisation of resources. First, research activities mainly take place in universities. Considering the difficulty of job promotion and approval of scientific research funds, scientific research teams in colleges and universities usually prefer applied research topics, which leads to the allocation of scientific research resources more inclined to these fields. In such circumstances, to strengthen academic communication and teamwork, basic research activities within universities are usually closely linked with related applied research, which is conducive to playing the role of basic research in promoting applied research to absorb and digest new technologies. The transformation from basic research results to applied value has become more efficient; however, the human capital cultivated by the scientific research team in the process of tackling key problems, especially the human capital of students, has strong mobility and knowledge spill-over effects, which can form the effective complementarity of various types of innovative knowledge and is favourable to the efficient operation of the regional economy. Second, more basic research activities have brought about the growth of the stock of basic knowledge, thereby enriching the technological opportunities and various heterogeneous knowledge of enterprises, and improving the technological level of enterprises and optimizing both the input-output ratio and economic performance. The theory of technological possibility, which advocates that the innovation behaviour of enterprises is dominated by the supply side, believes that technological opportunities are a more important factor than market demand in determining the technological progress of enterprises. The output of the basic research results helps enterprises acquire heterogeneous technical knowledge. Compared to the "imitation absorption model", it can break the original technology iteration route and guide the "deviant" innovation of enterprise products. After some companies use regional basic research input resources to

gain a first-mover advantage in technology, others in the industry will actively link to basic research input entities in the region and achieve rapid knowledge spill-over and industrial technological progress through large-scale use.

**Hypothesis II-1:** Human capital improvement mediates the effect of basic research investment on economic growth efficiency.

**Hypothesis II-2:** An increase in knowledge reserves is the mediating factor of the impact of basic research investment on the efficiency of economic growth.

## 4. Research design

### 4.1 Measurement model setting

Based on the theoretical analysis above, the impact of basic research investment on regional economic growth efficiency is investigated and the relevant mechanisms are tested. In line with the purpose of the empirical research, this study sets the benchmark model as follows:

$$tfp_{it} = \alpha_0 + \alpha_1 b\_rd_{it} + \alpha_2 ind_{it} + \alpha_3 urb_{it} + \alpha_4 mar_{it} + \mu_i + v_t + \varepsilon_{it} \qquad (1)$$

Among them, $tfp_{it}$ and $b\_rd_{it}$ are the explained variable economic growth efficiency and the explanatory variable basic research investment, respectively; this study introduces the industrial structure ($ind_{it}$), marketization degree ($mar_{it}$), and urbanisation level ($urb_{it}$) as control variables; $\mu_i$ is an individual dummy variable, $v_t$ is a time dummy variable, and $\varepsilon_{it}$ is a random perturbation term.

As mentioned above, as basic research investment has two impacts—promoting and inhibiting regional economic growth efficiency—there will be a quadratic functional relationship between basic research and regional economic efficiency. Therefore, this study introduces the square term of basic research investment into the model. Consequently, the basic model set above is further modified to:

$$tfp_{it} = \alpha_0 + \alpha_1 b\_rd_{it} + \alpha_2 b\_rd_{it}^2 + \alpha_3 ind_{it} + \alpha_4 urb_{it} + \alpha_5 mar_{it} + \mu_i + v_t + \varepsilon_{it} \qquad (2)$$

In the theoretical analysis part of this study, it is pointed out that an increase in basic research investment can improve the efficiency of regional economic growth by improving the level of human capital and increasing knowledge reserves. To test this mechanism, this study selects human capital($edu_{it}$) and knowledge reserves($ipa_{it}$) as mediating variables, draws on the mediating effect test method of Baron and Kenny and Wen Zhonglin, and constructs a three-step recursive econometric model to conduct the transmission mechanism [31, 32].

$$tfp_{it} = \Delta_0 + \theta_0 b\_rd_{it} + \theta_1 b\_rd_{it}^2 + q_0 X_{it} + \xi_{it} \qquad (3)$$

$$edu_{it} \, (ipa_{it}) = \Delta_1 + \theta_3 b\_rd_{it} + \theta_4 b\_rd_{it}^2 + q_0 X_{it} + \zeta_{it} \qquad (4)$$

$$tfp_{it} = \Delta_3 + \theta_5 b\_rd_{it} + \theta_6 b\_rd_{it}^2 + \theta_7 edu_{it} \, (ipa_{it}) + q_0 X_{it} + \xi_{it} \qquad (5)$$

Among them, $\Delta_0, \Delta_1$ and $\Delta_3$ are constant terms, $X_{it}$ represents the set of control variables, and $\zeta_{it}$ and $\xi_{it}$ are the random errors. First, Eq (3) is used to test the impact of basic research investment on the effect of economic growth. Second, Eq (4) is regressed to test whether the impact of basic research investment on the intermediary variable is significantly positive. The investment in basic research has significantly strengthened the regional human capital (knowledge reserve). Finally, a quantitative test of Formula (5) is carried out. If coefficients $\theta_5$, $\theta_6$ and $\theta_7$ are all significant, and coefficients $\theta_5$ and $\theta_6$ are lower than those of $\theta_0$ and $\theta_1$, it means that human capital improvement (increase in knowledge reserves) plays a role as a partial mediator in the process of basic research investment affecting the efficiency of economic growth. If the

two coefficients $\theta_5$ and $\theta_6$ are not significant, but $\theta_7$ is significant, it means that the improvement in human capital (increase in knowledge reserves) has a complete mediating effect in the process of basic research investment affecting the efficiency of economic growth.

## 4.2 Data source and variable description

The samples for the empirical study were selected from 31 provinces in China (autonomous regions and municipalities). The data were obtained from the "China Statistical Yearbook", "China Science and Technology Statistical Yearbook", "China Labor Statistics Yearbook", and National Bureau of Statistics, and was supplemented by linear interpolation. A moderate frontier technology gap is an important condition for basic research investments to promote economic growth [21]. In 2009, China's per capita national income and per capita GDP exceeded 3,000 dollars, and reached the forefront of middle-income countries according to the standards of the United Nations and the World Bank, thereby indicating that the gap between China and the world's frontier level was shrinking. Accordingly, this study chose 2009 as the time base to explore the impact of the proportion of basic research investment on regional economic growth. Considering that the relevant data in the "China Science and Technology Statistical Yearbook" have been updated up to 2019, this study selects the relevant data from 2009–2019 for empirical analysis.

1. Explained variable: economic growth efficiency (tfp). Currently, academic circles have formed three types of measurement methods for the measurement of economic growth efficiency. The first type is the total factor productivity based on various data envelopment methods. For example, Meng Wangsheng and Fangqin used real GDP as the expected output and industrial sulphur dioxide as the undesired output, and adopted a non-radial directional distance function model [33]. Ye Xiangsong employed the super-efficiency data envelopment method to measure the tfp of the Pearl River Delta region [34]. The second type of measurement method represents the efficiency of economic growth by constructing multidimensional comprehensive indicators. Only a few scholars use this method for measurement, such as Huang Fanhua and Guo Weijun [35]. The economic growth efficiency index is constructed from the three dimensions of productivity, labour productivity, capital productivity, and total factors, and the three sub-indices are synthesised using the entropy method. Finally, the economic growth efficiency index of each urban agglomeration in the Yangtze River Delta is formed. Fu Yuanhai et al. states that economic growth efficiency is reflected in the intensification level of economic growth and the input-output ratio. The third method is to directly measure the input-output efficiency of the economy through DEA, SFA, Super-SBM, and other methods and use it as a proxy variable for regional economic growth efficiency [36–38]. This study uses stochastic frontier analysis (SFA) to measure total factor productivity to represent regional economic efficiency, and the Solow residual method, DEA, and other methods to re-measure total factor productivity for a robustness test. The specific form of the SFA production function set in this study is as follows:

$$Y_{it} = \beta_0 + \beta_1 t + \beta_3 lnK_{it} + \beta_4 lnL_{it} + v_{it} - u_{it} \tag{6}$$

$$u_{it} = u_i exp[-\eta(t - T)] \tag{7}$$

$$\gamma = \frac{\sigma_u^2}{\sigma_u^2 + \sigma_v^2} \quad (0 \leq \gamma \leq 1) \tag{8}$$

where Y is the regional gross output value, which is deflated by the GDP deflator; K is the regional capital investment, which is estimated by the perpetual inventory method for

reference by Zhang Jun [39]: $K_{j,t} = (1−\delta)K_{j,t−1}+I_{j,t}$. Taking (1952) as the base period, the depreciation rate is set at 9.6%; L is the labour input, expressed by the number of employed persons in urban units at the end of the year. $v$ is a random error term, and has $v_i \sim iid\ N\ (0, \sigma_v^2)$. μ is the technical loss error term for computing technical inefficiency, and $u \sim N^+\ (\mu,\ \sigma_\mu^2)$; αvδ γ is the proportion of technical invalidity in the random disturbance term, which is estimated using the maximum likelihood method.

$$tfp_{it} = te_{it}\exp\ (\beta_0 + \beta_1 t) \tag{9}$$

$$te_{it} = \exp\ (−u_{it}) \tag{10}$$

2. Explanatory variables: Basic research input (b_rd). In this study, the basic research funds for R&D expenditures are expressed as a percentage of R&D expenditures.

3. Control variables: To avoid the error of omitted variables as much as possible, this study also controls for the following variables: (1) Industrial structure (ind), expressed as the proportion of the added value of the secondary industry to GDP. (2) Market mechanism (mar), expressed as the proportion of urban private and individual employment in urban unit employment (number at the end of the year). (3) Urbanisation level (urb), expressed as the proportion of the urban population in each region. (4) Labour cost (wage), expressed as the average wage of urban units.

4. Threshold variable: The theoretical mechanism analysis above shows that as the gap between the technological level and frontier technology narrows, the increase in investment in basic research will play a greater role in improving the efficiency of regional economic growth. In this study, the per capita gross domestic product (pergdp) is used as the threshold variable to investigate whether the increase in the proportion of basic research investment has played a greater role in promoting the improvement of economic efficiency with the continuous improvement of the regional economic level.

5. Mediating variables: ① Human capital (edu). Referring to Yu Binbin and Wu Yinzhong [40], this study uses the number of years of education per capita, which is calculated by multiplying the number of years of education at each educational level by the proportion of the total population. Illiteracy is set as 0 years, primary school as 6 years, 9 years for junior high school, 12 years for high school and secondary school, and 16 years for junior college or above. The specific calculation formula is as follows:

$$edu_{it} = \sum\nolimits_{k=1}^{5} W_k\theta_k \tag{11}$$

② Knowledge Reserve (ipa). Drawing on the processing methods of Yan Chengliang and Gong Liutang, Potter, and Pessoa [41–43], this study estimates the total knowledge stock in the economy based on the number of patent applications. The specific calculation formula is as follows. The depreciation rate refers to the practice of Deng Ming and Qian Zhengming, takes d = 0.0714, and calculates with 2009 as the base period [44].

$$A_t = (1 − d)A_{t−1} + P_{t−1} \tag{12}$$

$$A_0 = \frac{P_0}{g + d} \tag{13}$$

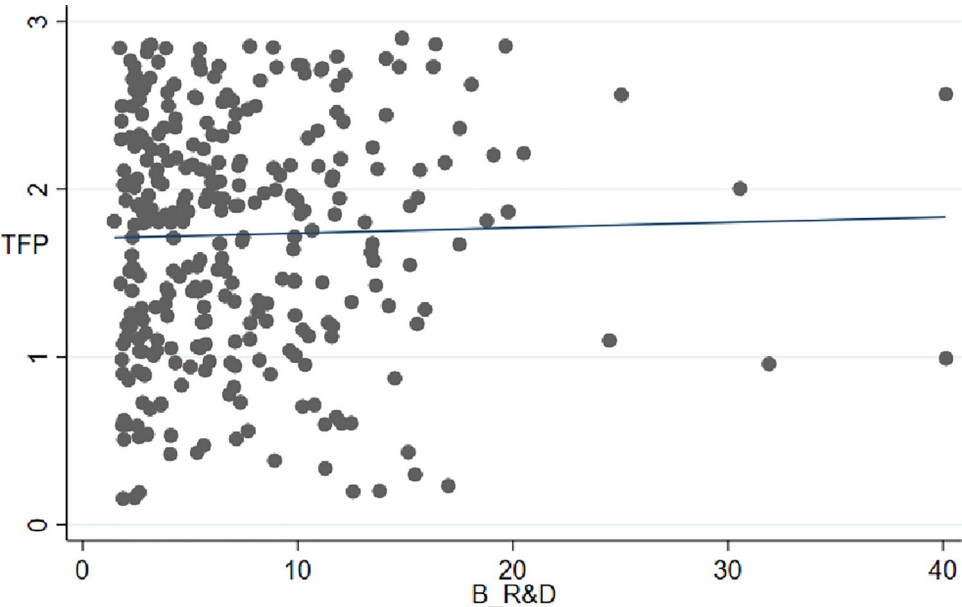

**Fig 1. Scatter plot of basic research investment and regional economic growth efficiency.**

## 4.3 Analysis on characteristic facts

Before the quantitative analysis, it is necessary to carry out a preliminary statistical analysis of the empirical data that has been mastered. Fig 1 presents the scatter analysis results for basic research investment and regional economic growth efficiency. There is a statistically significant positive correlation between basic research investment and regional economic growth from the data fitting situation. The fitting results of the quadratic relationship of the efficiencies are better.

## 5. Results and discussion

Table 1 reports descriptive statistical results for the main variables. The average proportion of basic research investment is 5.62%, which is far lower than the general level for developed countries (the proportion for developed countries such as the United States, Britain and France is between 12% and 23%, and the United States is at 15%). Based on the Hausman test

**Table 1. Descriptive statistics.**

| variable | N | min | p50 | mean | max | sd |
|---|---|---|---|---|---|---|
| tfp | 341 | 0.160 | 1.710 | 1.640 | 2.900 | 0.730 |
| b_rd | 341 | 1.430 | 5.620 | 7.170 | 40.13 | 5.540 |
| ind | 341 | 0.160 | 0.460 | 0.440 | 0.590 | 0.0900 |
| urb | 341 | 0.220 | 0.540 | 0.550 | 0.900 | 0.140 |
| mar | 341 | 0.300 | 0.940 | 1.040 | 2.980 | 0.440 |
| wage | 341 | 24165 | 53426 | 57391 | 170000 | 23134 |
| pergdp | 341 | 10309 | 43223 | 49841 | 160000 | 26328 |
| edu | 341 | 4.160 | 9.850 | 9.790 | 13.83 | 1.270 |
| ipa | 341 | 5.320 | 10.56 | 10.32 | 13.74 | 1.660 |
| solow_tfp | 341 | -0.820 | 0.0300 | 0 | 0.850 | 0.250 |
| rd_human | 341 | 0.0200 | 0.0900 | 0.110 | 0.400 | 0.0700 |

**Table 2. Diagnostic tests results.**

|  | Statistics | p-value |
|---|---|---|
| Hausman test | 13.44 | 0.0195 |
| Boostrap Hausman test | 13.76 | 0.0324 |
| Modified Wald test | 25.94 | 0.7243 |

and the Boostrap Hausman test in Table 2, the fixed-effects model and the random-effects model are judged, and the fixed-effects model is finally selected based on the test results. Further, an individual fixed-effects model is selected by the F-test results and the economic significance of the model itself. Then, the Modified Wald test is used to determine whether the model has an intergroup heteroscedasticity problem. The test results show that there is heteroscedasticity among groups, so the robustness standard error is used in the model estimation to overcome it.

## 5.1 Benchmark results analysis

Table 3 reports the estimated results for the impact of basic research investment on the efficiency of regional economic growth. By gradually introducing control variables to verify the impact of basic research investment on economic growth effects, it is found that the sign and significance of the core explanatory variables and control variables do not change significantly, which indicates that the fixed effect estimation results show strong robustness. From the regression results of models (1)—(5), the coefficients of basic research investment are positive and pass the significance test at the level of 5%, thus indicating that the increase in the proportion of basic research investment contributes to the improvement of regional economic growth efficiency, which is consistent with the research conclusions of Yan and Gong [41]. The coefficient of the core explanatory variable has not changed in models (1)—(5), thus suggesting that the empirical results are robust to some extent. Our conclusion can also be demonstrated by international experience [45]. Chen considered the dynamic relationship between R&D investment and total factor productivity under financial constraints, and tried to explore to what extent the R&D channel explained the productivity loss caused by financial constraints [46]. Tsamadias et al. (2018) investigated the interaction between R&D (R&D), human capital (HC), foreign direct investment (FDI) and total factor productivity (TFP) in OECD countries from 1995 to 2015, and found in their empirical study that R&D investment can positively improve TFP [47]. Wang also reached a similar conclusion in the field of agricultural research, but their study believed that the positive impact of increasing R&D activities on productivity needs to be played in the long run [48].

Meanwhile, the regression coefficients of the quadratic terms of the proportion of investment in basic research ($b\_rd^2$) are all negative, and they have passed the significance test at the 5% level, which indicates that the proportion of investment in basic research has an "inverted U-shaped" effect on the regional economic growth efficiency path. Thus, Hypothesis I is verified. This is an important conclusion found in this paper. The possible reasons for this path of influence are that, on the one hand, an appropriate increase in the proportion of basic research investment can optimize the efficiency of scientific research resource allocation, and achieve high-efficiency growth of the regional economy by improving the ability of industrial independent innovation. On the other hand, if the proportion of basic research investment is too high, it may lead to an imbalance between basic research and applied research, affect the efficiency of the whole scientific research system from "knowledge technology to productivity", and eventually deteriorate the efficiency of regional economic growth.

**Table 3. Benchmark regression results.**

|  | (1) | (2) | (3) | (4) | (5) | (6) |
|---|---|---|---|---|---|---|
|  | tfp | tfp | tfp | tfp | tfp | tfp |
| b_rd | 0.1154*** | 0.1105** | 0.1169** | 0.1094** | 0.1078** |  |
|  | (0.0353) | (0.0421) | (0.0461) | (0.0463) | (0.0462) |  |
| b_rd$^2$ | -0.0021*** | -0.0020** | -0.0024** | -0.0025** | -0.0025** |  |
|  | (0.0007) | (0.0008) | (0.0009) | (0.0009) | (0.0009) |  |
| ind |  | -0.2820 | 2.6970** | 3.3001*** | 3.3866*** | 2.5698** |
|  |  | (0.6601) | (1.1215) | (1.1431) | (1.2139) | (1.1893) |
| urb |  |  | 5.4002*** | 3.0484** | 2.7224 | 6.2194*** |
|  |  |  | (0.9245) | (1.2334) | (1.8239) | (2.4131) |
| mar |  |  |  | 0.5388*** | 0.5299*** | 0.5045*** |
|  |  |  |  | (0.1593) | (0.1592) | (0.1914) |
| wage |  |  |  |  | -1.18e-06 | 8.52e-06 |
|  |  |  |  |  | (4.86e-06) | (5.71e-06) |
| _cat#c.b_rd 0 |  |  |  |  |  | 0.0149 |
|  |  |  |  |  |  | (0.0167) |
| _cat#c.b_rd 1 |  |  |  |  |  | 0.1122*** |
|  |  |  |  |  |  | (0.0394) |
| cons | 0.9842*** | 1.1362** | -3.1966*** | -2.6583** | -2.4287* | -4.1030** |
|  | (0.1962) | (0.4855) | (1.1116) | (1.1174) | (1.4130) | (1.7313) |
| Obs | 341 | 341 | 341 | 341 | 341 | 341 |
| R$^2$ | 0.0285 | 0.0287 | 0.0757 | 0.0997 | 0.0999 | 0.1070 |

Note:

*** express $p<0.01$

** express $p<0.05$

* express $p<0.1$; Standard errors are in parentheses, the same below

In terms of control variables, the influence coefficient of industrial structure (ind) on the efficiency of economic growth is positive and has passed the significance test at the 1% level, which indicates that an increase in the proportion of industrial structure has a positive impact on the efficiency of regional economic growth. The possible explanations are as follows. According to Baumol's cost disease theory, the service industry has a lower labour productivity compared with industry. When the regional industrial structure evolves to the tertiary industry, it is easy to cause the economy to fall into a "structural slowdown". The regression coefficient of urbanisation level (urb) is positive and has passed the significance test at the 1% level, which indicates that the improvement in urbanisation level can significantly improve the efficiency of regional economic growth. The reason for this phenomenon is that as China's urbanization process switches from the investment driven and government led mode to a new stage in which the market determines the allocation of resources, a higher level of urbanization provides a greater competitive market and evolutionary incentives for enterprise agglomeration, market competition, and technical cooperation. The influence coefficient of marketization level (mar) on economic growth efficiency is positive and highly significant. Overall, these are expected findings as most of the empirical literature suggests. The regression coefficient for labour level (wage) is positive but does not pass the significance test.

In terms of the threshold effect test of the impact of basic research investment on economic growth efficiency, the Bootstrap method was used to make 300 repeated sampling judgments for the single, double, and triple thresholds of the model. The test results are listed in Table 4,

**Table 4. Threshold sampling results.**

|  | Threshold number | Threshold value | BS frequency | RSS | F value | P value |
|---|---|---|---|---|---|---|
| pergdp | single threshold | 115053.0000 | 300 | 161.5457 | 9.00 | 0.0367 |
|  | double threshold | 94648.0000 | 300 | 158.6701 | 5.98 | 0.2733 |
|  | three thresholds | 80932.0000 | 300 | 157.0086 | 3.49 | 0.9467 |

and it can be seen that there is a threshold effect of a single threshold. Model (6) in Table 3 provides specific test results for the single-threshold effect. It can be found that when the per capita national income does not cross the threshold, the estimated sign of the parameter of the proportion of investment in basic research is positive but fails the significance test. When the per capita national income level crosses the threshold, the parameter estimation sign of the proportion of investment in basic research is positive and passes the 1% significance level test. This estimation shows that with the evolution of the development stage, basic research investment will play a more significant role in improving the efficiency of economic growth.

## 5.2 Mediating effect test

Table 5 reports the test results of the intermediary mechanism of the proportion of investment in basic research affecting the efficiency of regional economic growth. As can be seen from Table 5, the results of model (2) reveal that basic research investment significantly improves the level of human capital and presents an inverted U-shaped impact path. The parameter estimation results of human capital in model (3) are significantly positive at the 1% level, and the coefficient values of basic research and its quadratic terms are significantly lower than those of model (1), which means that investment in basic research affects the regional economy. In the process of increasing efficiency, the improvement of human capital has played a part in the mediating effect, and hypothesis II-1 has thus been verified. It is mainly because, on the one hand, the enhancement of human capital level can promote micro level enterprises to improve their knowledge absorption and digestion capacity and organizational operation efficiency in the process of technological progress. On the other hand, it can realize the external explicit technology popularization and tacit knowledge diffusion in the process of self-study through the cross enterprise flow between foreign direct investment and local enterprises, and accelerate the application of technology industrialization to advance the overall production efficiency of the industry.

The results of Model (4) establish that basic research investment significantly increases knowledge reserves and presents an inverted U-shaped impact path. This study measures the level of regional knowledge stock based on the number of regional patent applications. The process of patent application and grants involves both theories and knowledge summarized in basic research, and the development, application, and reorganization of new perspectives in applied research. The excess basic research investment will inhibit the development of applied research, and ultimately reduce the growth of regional knowledge reserves, which is manifested as a U-shaped action path that promotes first and then inhibits. In Model (5), the parameter estimation result of increasing knowledge reserves is negative, but it fails the significance test. This result means that basic research affects the efficiency of regional economic growth in the process of investment. The role of an intermediary factor played by the increase in knowledge reserves has not been supported by empirical data. A possible reason is that, although the importance of basic research to original innovation and industrial progress has been paid increasing attention, low-quality patents and knowledge based on imitation innovation still occupy the overall knowledge reserve. However, due to the long-term separation

**Table 5. Test results of human capital and knowledge reserves as an intermediary factor.**

|  | (1) | (2) | (3) | (4) | (5) |
|---|---|---|---|---|---|
|  | tfp | edu | tfp | ipa | tfp |
| b_rd | 0.1078** | 0.0421*** | 0.0908* | .01944* | 0.1043** |
|  | (0.0462) | (0.0148) | (0.0467) | (.01142) | (0.0466) |
| b_rd$^2$ | -0.0025** | -0.0018*** | -0.0017* | -0.00094*** | -0.0023** |
|  | (0.0001) | (0.0003) | (0.0009) | (0.0002) | (0.0009) |
| edu |  |  | 0.4044*** |  |  |
|  |  |  | (0.0688) |  |  |
| ipa |  |  |  |  | 0.1838 |
|  |  |  |  |  | (0.1822) |
| ind | 3.3866*** | 0.6840 | 3.1100** | 0.3164 | 3.3285** |
|  | (1.2139) | (0.7640) | (1.1521) | (0.5073) | (1.2113) |
| urb | 2.7224 | 3.4503*** | 1.3271 | 8.1770*** | 1.2198 |
|  | (1.8239) | (1.0345) | (1.7793) | (0.9679) | (2.1455) |
| mar | 0.5299*** | 0.0496 | 0.5099*** | 0.0753 | 0.5161*** |
|  | (0.1592) | (0.0910) | (0.1718) | (0.0893) | (0.1577) |
| wage | -1.18e-06 | -0.00002*** | 6.56e-06 | -1.63e-05*** | 1.82e-06 |
|  | (4.86e-06) | (2.41e-06) | (5.34e-06) | (1.91e-06) | (6.63e-06) |
| cons | -2.4287* | 8.4679*** | -5.8531*** | 6.4426*** | -3.6125* |
|  | (1.4130) | (0.7735) | (1.7377) | (0.6590) | (2.0486) |
| Obs | 341 | 341 | 341 | 341 | 341 |
| R$^2$ | 0.0999 | 0.6790 | 0.1302 | 0.9034 | 0.1023 |

between the production system and R&D activities, the transformation cycle of heterogeneous knowledge from output to productivity application is long, which can usually only be observed in the long run. The driving effect of knowledge production on economic growth cannot reflect the impact of the growth in relevant knowledge reserves on economic efficiency in the current period.

## 5.3 Heterogeneity discussion

Referring to the practice of Yu and Shen [49], by introducing regional dummy variables and constructing the interaction terms between the dummy variables and core explanatory variables, we analyse the impact of basic research investment in different regions on the efficiency of economic growth. Models (1) and (2) in Table 6 reveal that the proportion of basic research investment in the eastern and central regions and regional economic growth efficiency show an "inverted U-shaped" path of action. The industrial structure and technical level of the eastern and central regions are above the national average level, especially in the eastern region, as the region with the most intensive regional economic activities in China has gathered numerous high-tech enterprises, advanced manufacturing, and high-end productive service industries. It has a moderate frontier gap and a strong willingness to innovate. Therefore, basic research investment can promote the industry's TFP by strengthening the independent innovation capabilities of micro-enterprises and generating positive R&D externalities, thereby improving the efficiency of regional economic growth. The estimation results of Model (3) in Table 6 indicate that the regression coefficient of the proportion of investment in basic research in the western region is negative, but it does not pass the significance test. It implies that for the western region, an increase in the proportion of investment in basic research

**Table 6. Heterogeneity test results.**

| | East Midlands | | Western Region | | tfp |
|---|---|---|---|---|---|
| | (1) | (2) | (3) | (4) | (5) |
| Region×b_rd | 0.0710** | 0.2061*** | -0.0171 | 0.0910 | |
| | (0.0324) | (0.0621) | (0.0437) | (0.1506) | |
| Region×b_rd$^2$ | | -0.0045** | | -0.0053 | |
| | | (0.0016) | | (0.0060) | |
| b_rd | | | | | -0.0957*** |
| | | | | | (0.0295) |
| b_rd$^2$ | | | | | -0.0010** |
| | | | | | (0.0004) |
| tfp×b_rd | | | | | 0.0813*** |
| | | | | | (0.0135) |
| ind | 3.4035*** | 4.0569*** | 2.5886** | 2.4815** | 1.2531* |
| | (1.0761) | (1.1927) | (1.0963) | (1.0516) | (0.6956) |
| urb | 3.0890 | 3.67317* | 2.1980 | 2.2286 | 1.5379* |
| | (1.9775) | (2.0647) | (1.6853) | (1.6813) | (0.8680) |
| mar | 0.5282*** | 0.5268*** | 0.5380*** | 0.5242*** | 0.0871 |
| | (0.1510) | (0.1554) | (0.1547) | (0.1560) | (0.1124) |
| wage | -4.92e-07 | 2.51e-07 | -2.30e-06 | -2.23e-06 | -9.62e-07 |
| | (5.26e-06) | (5.45e-06) | (4.73e-06) | (4.68e-06) | (1.67e-06) |
| cons | -2.3623 | -3.3521** | -1.1078 | -1.2223 | -0.0039 |
| | (1.4554) | (1.5635) | (1.1861) | (1.1981) | (0.7714) |
| Obs | 341 | 341 | 341 | 341 | 341 |
| R$^2$ | 0.1026 | 0.1133 | 0.0822 | 0.0838 | 0.6365 |

cannot promote the efficiency of regional economic growth. A possible reason is that most of the economic development in the western region relies on natural resources and the transfer of industries in the eastern and central regions. The corresponding production technology level is far behind the world's cutting-edge technologies. It is more suitable to adopt an R&D resource allocation pattern of "emphasizing application and ignoring foundation". The increase in investment in basic research makes it difficult to support the upgrade and optimisation of the regional industrial structure, and this fact is reflected in the crowding out of applied research resources. The estimation results of model (4) demonstrate that investment in basic research in the western region has not yet formed a U-shaped relationship.

There may be a circular cumulative effect between basic research investment and regional technological progress and economic growth, which leads to the continuous optimization of industrial structure and the strengthening of industry university research cooperation in regions with high economic growth efficiency. Subsequently, the increase in the proportion of basic research investment will play a greater role in promoting the improvement of economic growth efficiency. To verify the possible heterogeneity, this study introduces the interactive term between economic growth efficiency and core explanatory variables to conduct an empirical test. The estimation results of Model (5) in Table 6 reveals that the multiplication term of economic growth efficiency and core explanatory variables is positive and has passed the significance test at the 1% level. It suggests that compared to regions with low economic efficiency, basic research investment can play a greater role in promoting economic growth in regions with higher economic growth efficiency; that is, there is a "self-reinforcing" mechanism between the two.

**Table 7. Robustness test results.**

|  | (1) | (2) |
|---|---|---|
|  | **solow_tfp** | **tfp** |
| b_rd | 0.03639** |  |
|  | (0.01485) |  |
| b_rd$^2$ | -0.0006** |  |
|  | (0.00029) |  |
| rd_human |  | 13.74075*** |
|  |  | (4.80635) |
| rd_human$^2$ |  | -23.83499* |
|  |  | (12.73865) |
| ind | -1.72343*** | 3.78404*** |
|  | (0.49572) | (1.1974) |
| urb | -0.55251 | 2.84527 |
|  | (0.64188) | (1.79191) |
| mar | 0.17864** | 0.50752*** |
|  | (0.07389) | (0.15389) |
| wage | 0.00001*** | -8.07e-07 |
|  | (1.29e-06) | (4.81e-06) |
| cons | 0.33059 | -3.20343** |
|  | (0.46437) | (1.33754) |
| Obs | 341 | 341 |
| R$^2$ | 0.19771 | 0.10875 |

## 5.4 Robustness test

To test the reliability of the above conclusion on the effect of basic research input on regional economic growth efficiency, this study conducts the following robustness tests: (1) re-measure regional economic growth efficiency (solow_tfp) using the Solow residual value method and an empirical test; and (2) change the measurement method of the explanatory variables and replace the core explanatory variables with the index of the number of R&D personnel invested in basic research in the total R&D personnel (rd_human). According to the regression results in Table 7, the sign and significance of the core explanatory variables in Models (1)–(2) did not change significantly, thereby indicating that the benchmark regression results in this study are relatively robust.

## 6. Conclusions and implications

This study takes the current Chinese R&D resource allocation structure as the research object, and analyses the relationship between the proportion of basic research investment and economic growth efficiency. Although numberous studies have focused on R&D activities, productivity, human capital, knowledge stock and other aspects, the novelty of this work lies in the inclusion of the proportion of basic research investment, innovation capacity and regional economic growth efficiency in a single framework for analysis. Simultaneously, it conducts an empirical test on the relationship and its action path with the help of panel data from 31 provinces in the fixed effect model and the mediation effect test model from 2009 to 2019.

Above all, panel data empirical test shows that the increase in the proportion of basic research investment can improve the efficiency of regional economic growth, but there is an inverted U-shaped effect path that promotes first and then suppresses, which is supported by a series of subsequent robustness tests. The test results of the threshold model further point out

that, with the transition of economic development stages, investment in basic research will play a more significant role in promoting the efficiency of economic growth. The results of the intermediary mechanism test show that an increase in the proportion of investment in basic research can improve the efficiency of regional economic growth by increasing the level of human capital, which shows a partial intermediary effect. The increase in the proportion of investment in basic research has a promoting effect on the increase of regional knowledge reserves, but the path of increasing knowledge reserves has not been supported by empirical data, thus affecting the efficiency of regional economic growth. Heterogeneity test results reveal that the research conclusion in the eastern and central regions is consistent with the regression conclusion of the national sample, that is, the increase in the proportion of basic research investment can positively improve the regional economic growth efficiency and show an inverted U-shaped effect path. The west regional research conclusions are in complete contrast. The proportion of basic research investment has a negative impact on regional economic growth efficiency, but it does not pass the significance test.

These findings indicate that basic research input is a significant contributor to TFP growth. This result has important policy implications for latecomer economies close to the world's frontier production level.Moreover, when the economic development has passed a certain stage (after the per capita income has exceeded a certain critical value), the increase in the proportion of basic research investment has a greater positive impact. The above research conclusions are basically consistent with those in the existing literature [45, 50]. In addition, this conclusion also responds to Song's view that private BR investment in the digital era has a positive effect when studying the impact of basic research investment on enterprise performance at the micro level [51].

On the Chinese perspective, specific attention should be paid to the basic research projects. First, local governments should be aware of the core driving role of basic research activities for economic growth. They should not only pay attention to the growth of total R&D investment to seek new technological innovation and industrial catch-up but also ensure the optimization of resource allocation of R&D activities in the era of the knowledge economy. It is necessary to tap the innovation potential brought by resource optimization in R&D departments by increasing the proportion of basic research investment. Second, in the process of adjusting the resource allocation pattern of R&D activities, on the one hand, we should pay attention to the deployment of scientific research forces in emerging strategic industries. On the other hand, it is necessary to open up the links among enterprises, R&D institutions and universities, build a tripartite benefit sharing mechanism, and improve the efficiency of technological innovation. Third, in the context of accelerating the upgrading of industrial structure and breaking the blockade of technology sources, relevant departments must emphasize the scientific and cultural education in the region and strengthen the cultivation and accumulation of human capital. On this basis, by virtue of the corresponding talent welfare policies, we will attract high-quality labour into the region and support the efficient growth of the regional economy. Fourth, the eastern coastal areas of China need to speed up the adjustment of the allocation pattern of scientific research resources and form an independent core controllable leading industrial chain and R&D innovation chain by strengthening investment in basic research projects.

Although there are some innovations in the theoretical analysis framework and research perspective, there are three limitations and shortcomings: (1) the data only use China's panel data from 2009 to 2019, thus lacking data and empirical analysis from other economies. (2) The sample size of provincial level research is small, so future research should use more micro level data to support our conclusions. This study also lacks the test and analysis of the relationship between the two from the perspective of spatial correlation. (3) Although it investigates

the impact path of basic research investment on regional economic efficiency from the perspectives of human capital cultivation and basic knowledge enrichment, there may be other impact paths and intermediary variables. Simultaneously, the follow-up research can further build a mathematical analysis model to more comprehensively investigate and analyse the impact of the proportion of basic research investment on regional economic growth efficiency. Moreover, future research should assess the link between basic research investment proportion and total factor productivity by using more micro empirical database. In addition, due to the widespread spatial dependence among regions, subsequent studies can further introduce spatial econometric models to conduct more in-depth and detailed analysis of panel data.

## Supporting information

**S1 Data.**
(XLSX)

## Author Contributions

**Conceptualization:** Bin Xu.

**Data curation:** Yichen Jiang, Bin Xu.

**Formal analysis:** Yichen Jiang.

**Investigation:** Bin Xu.

**Methodology:** Wang Hui.

**Project administration:** Boyue Sun.

**Resources:** Boyue Sun, Liyan Hu.

**Software:** Yichen Jiang, Li Fang.

**Supervision:** Li Fang, Liyan Hu.

**Validation:** Li Fang, Wang Hui.

**Visualization:** Wang Hui.

**Writing – original draft:** Yichen Jiang.

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
