## [Decision Letter · Decision Letter 0]

7 Jun 2022

PONE-D-22-13395Basic Research Investment, Innovation Capability Improvement and Economic Growth EfficiencyPLOS ONE

Dear Dr. hui,

Thank you for submitting your manuscript to PLOS ONE. After careful consideration, we feel that it has merit but does not fully meet PLOS ONE’s publication criteria as it currently stands. Therefore, we invite you to submit a revised version of the manuscript that addresses the points raised during the review process.

The submission requires further revisions with reference to study novelty, prior literature, econometric setting and empirical findings’ interpretation, as well as policy implications. The submission requires further revisions with reference to study novelty, prior literature, econometric setting and empirical findings’ interpretation, as well as policy implications.

We look forward to receiving your revised manuscript.

Kind regards,

Stefan Cristian Gherghina, PhD. Habil.

Academic Editor

PLOS ONE

Journal Requirements:

2. In your Methods, please provide the exact URL to the data used in this study.

Reviewers' comments:

Reviewer's Responses to Questions

**Comments to the Author**

1. Is the manuscript technically sound, and do the data support the conclusions?

Reviewer #1: No

Reviewer #2: Yes

2. Has the statistical analysis been performed appropriately and rigorously? 

Reviewer #1: No

Reviewer #2: Yes

3. Have the authors made all data underlying the findings in their manuscript fully available?

Reviewer #1: No

Reviewer #2: Yes

4. Is the manuscript presented in an intelligible fashion and written in standard English?

Reviewer #1: No

Reviewer #2: Yes

5. Review Comments to the Author

Reviewer #1: The Abstract must report the aim of the study, the basic information on the sample (time span, countries analyzed), the empirical methodology used, the main findings, and the relevant policy implications.

Introduction and Literature Review should be split into two different sections.

The Introduction should highlight the relevance of the topic, the novelty of the results, the importance of policy implications, the sample’s choice, the methodology’s appropriateness, the data used, the contribution to the literature, and the limitations of the study.

The literature review is partial and incomplete, and some recent and relevant contributions should be cited and discussed: i.e., 10.1007/s40888-022-00261-z; 10.1016/j.jclepro.2021.129050; 10.1080/15567249.2020.1868622; 10.1016/j.jup.2021.101256.

The theoretical framework should be discussed more in detail.

The estimated model must be justified in light of the literature on this specific topic.

Descriptive statistics are absent.

Diagnostic tests are absent.

Robustness checks are absent.

The results should be discussed more in detail.

Comparisons with previous studies are absent.

Conclusions are too short.

Policy implications are weak.

Further research should be indicated.

Limitations of the study are not provided.

Proofreading by a native speaker is required.

The editing does not follow the journal’s guidelines.

The originality value of the study is limited.

This is a basic econometric exercise without a clear innovative intuition.

How does the paper enrich the knowledge of the scientific community?

Reviewer #2: Dear Authors,

The paper addressed an interesting topic in the context of Basic Research Investment, Innovation Capability Improvement and Economic Growth Efficiency. The theoretical section is quite acceptable but needs to discuss recent literature. The empirical section needs more theoretical discussion.

Below are some recommendations to enhance the quality of this research;

1. Abstract: should be written again in academic forms. (Main aim, where study conducted, model used, period, and main findings). Statements are missing.

2. Introduction: The author (s) should highlight the contribution of this work to the existing literature review in the last part of the introduction. The written contribution is not clear and the statements are too long.

3. Literature review: This needs to discuss recent literature to develop the theoretical argument and gaps.

4. Research Methods: reason behind the sample selection, period, and models used should be explained in detail. Which panel model been used? Diagnostic tests should be performed to confirm that the model is BLUE (best linear unbiased estimator), clear of heteroscedasticity and autocorrelation.

5. Empirical results and discussion: this section needs better discussion both statistically and theoretically.

6. Conclusion and policy implications: is fine.

Best of Luck

6. PLOS authors have the option to publish the peer review history of their article (what does this mean?). If published, this will include your full peer review and any attached files.

Reviewer #1: No

Reviewer #2: No

---

## [Author Response · Author response to Decision Letter 0]

29 Jul 2022

Dear Review Expert/Editorial Teacher,

Hello! Thank you very much for your constructive suggestions for revision of the study. Each of your review comments is an important help in revising and improving it. After receiving the revision opinions sent by your publication, the author immediately re-consulted the relevant literature and materials, and revised and supplemented according to the opinions of the review experts. The relevant changes are described as follows:

Reviewer #1

Q1:The Abstract must report the aim of the study, the basic information on the sample (time span, countries analyzed), the empirical methodology used, the main findings, and the relevant policy implications.

Modification instructions: The author attaches great importance to expert advice. According to experts' opinions, the author revises and adjusts the abstract greatly, including adding the research purpose of the study, highlighting the research data and methods, etc. The revised summary is below：

Abstract: This study explores the relationship between the proportion of basic research investment in R&D expenditure and regional economic growth efficiency and its influence path. A panel of data form 31 Chinese provinces was analysed by the fixed-effects model over 2009 to 2019. Empirical results highlight that: (1) The increase in the proportion of basic research investment has a significant promoting effect on regional economic growth efficiency, but it presents an "inverted U-shaped" impact path. Simultaneously, the threshold effect model test results reveal that basic research investment plays a more significant role in promotion when the per capita income level exceeds a certain threshold. (2) The test results of the mediating effect model indicate that the improvement of the level of human capital is the intermediate channel through which the investment in basic research affects regional economic growth efficiency, while the mediating effect of the increase of knowledge storage on the process of the investment in basic research affecting regional economic growth efficiency is not supported by data. The relevant policy implications are as follows: when the per capita income level of a region rises to a certain stage, the direction of research development should be changed from attaching importance to applied research and foreign technology import to coordinating applied research and basic research, to form independent innovation ability and improve the efficiency of regional economic growth.

Q2:Introduction and Literature Review should be split into two different sections.

Modification instructions: The author has re-written the introduction and literature review in two parts as prompted by the experts.

Q3:The Introduction should highlight the relevance of the topic, the novelty of the results, the importance of policy implications, the sample’s choice, the methodology’s appropriateness, the data used, the contribution to the literature, and the limitations of the study.

Modification instructions: We are very grateful to the anonymous reviewers for their comments. The author has made major changes to the introduction in accordance with the expert suggestions. The revised results of the introduction are as follows:

Since its reform and subsequent opening up, China's economy has developed rapidly and has achieved the "China Miracle", which has attracted global attention. In this process, the "introduction, absorption, and re-digestion" model, represented by imported equipment, foreign direct investment (FDI), and purchase of foreign technology, has played an important role in improving China's technological level and ensuring rapid economic growth. However, as the gap between China's economic development and the world's frontier level continues to narrow, the latecomer advantage that could have been earned from the experience gained and lost through the industrial, technological, and economic advancement of developed countries has gradually disappeared. From the perspective of the external environment, developed economies strictly limit the spill-over of high-technology and advanced knowledge, retain high-end productive service industries in the process of international industrial division, outsource non-core businesses, and firmly grasp the key technologies in the industrial chain and the stagnant core technologies. This situation means that it is increasingly difficult to directly absorb the technology and knowledge of developed economies. From the perspective of the internal environment, China's economy has entered a new stage, and the previous simple technology import mode is beginning to appear "powerless" in the face of pain points in the process of technological development. As the source of all scientific and technological innovation activities, basic research is not only the breakthrough point of various technical problems but also related to China's overall science and technology strategy. The 19th National Congress of the Communist Party of China pointed out that it is necessary to aim for the forefront of global scientific and technological development and to achieve breakthroughs in original achievements by strengthening basic research, to provide strong support for strategies such as building a strong country in science and technology and in quality. Once China's economic development has entered a new stage, whether the traditional scientific research resource allocation pattern of "emphasising application research but ignoring basic research" can adapt to the situation of economic development and improve total factor productivity is a question worthy of careful consideration.

The possible marginal contributions of this study are as follows. First, from the research perspective, unlike the existing literature that focuses on the overall R&D activities or the total amount of basic research investment, this study attempts to explore the factors that may improve the efficiency of regional economic growth from the structural perspective of R&D resource investment. Second, in terms of theoretical analysis, it constructs an analytical framework between the proportion of basic research investment in R&D activities and the efficiency of regional economic growth. Taking China, a typical representative of developing countries, as the research sample, the provincial level data of China from 2009 to 2019 are selected for empirical tests. Finally, in the empirical analysis, this study comprehensively uses the intermediary effect model, threshold effect model, and other empirical methods to carefully test the intermediary channels and various heterogeneous effects on the proportion of basic research investment that affects the efficiency of regional economic growth.

Q4:The literature review is partial and incomplete, and some recent and relevant contributions should be cited and discussed:i.e., 10.1007/s40888-022-00261-z;10.1016/j.jclepro.2021.129050; 10.1080/15567249.2020.1868622; 10.1016/j.jup.2021.101256.

Modification instructions: the author agrees with the expert opinion very much. Therefore, the author adds the literature review section, and discusses the latest relevant literatures, such as 10.1016/j.jclepro.2021.129050; 10.1080/15567249.2020.1868622 and others to develop the theoretical argument and gaps. In view of the excessive revision of the literature review section, the author suggests experts to check the modification results in Revised Manuscript with Track Changes.

Q5:The theoretical framework should be discussed more in detail.

Modification instructions: After carefully reading the theoretical mechanism section, the author very agreed with the comments made by the reviewers. So, the author has made major changes to the theoretical mechanism. Several details of the theoretical mechanism section are supplemented and enriched. Due to the excessive and lengthy revisions, the author suggests that experts can go to the original text to view the the changes.

Q6:The estimated model must be justified in light of the literature on this specific topic.

Modification instructions: The author refers to the practices of relevant literature, such as the empirical analysis method of Li et al. (2020) on the impact of basic research on technological progress, and the empirical analysis of Yan and Gong (2013) on the R&D driven economic growth model. On this basis, combined with the econometric model test method, this study finally determines to use the fixed effect model for empirical analysis.

Q7:Descriptive statistics are absent.

Modification instructions: The author attaches great importance to the opinions of experts. The authors have re-added a descriptive statistical analysis table of the main variables to the research. Experts can review the revised results of the paper in the original text.

Table 2 Descriptive statistics

variable N min p50 mean max sd

tfp 341 0.160 1.710 1.640 2.900 0.730

b_rd 341 1.430 5.620 7.170 40.13 5.540

ind 341 0.160 0.460 0.440 0.590 0.0900

urb 341 0.220 0.540 0.550 0.900 0.140

mar 341 0.300 0.940 1.040 2.980 0.440

wage 341 24165 53426 57391 170000 23134

edu 341 4.160 9.850 9.790 13.83 1.270

inn_ability 341 0.0300 0.200 0.240 0.760 0.120

dea_tfp 330 0.350 0.850 0.850 1.550 0.200

solow_tfp 341 -0.820 0.0300 0 0.850 0.250

rd_human 341 0.0200 0.0900 0.110 0.400 0.0700

pergdp 341 10309 43223 49841 160000 26328

Q8:Diagnostic tests are absent.

Modification instructions: Thanks to the experts' opinions, the authors have added econometric diagnostic tests related to model selection in the study. At the same time, the authors also added corresponding explanations. The result of the modification is as follows:

Based on the Hausman test and the Boostrap Hausman test, the fixed-effects model and the random-effects model are judged, and the fixed-effects model is finally selected based on the test results. Further, an individual fixed-effects model is selected by the F-test results and the economic significance of the model itself. Then, the Modified Wald test is used to determine whether the model has an intergroup heteroscedasticity problem. The test results show that there is heteroscedasticity among groups, so the robustness standard error is used in the model estimation to overcome it.

Table 3 Diagnostic tests results

 Statistics p-value

Hausman test 13.44 0.0195

Boostrap Hausman test 13.76 0.0324

Modified Wald test 25.94 0.7243

Q9:Robustness checks are absent.

Modification instructions: Thank you very much for the valuable comments from the anonymous reviewers. The authors re-added the results of the robustness test section. It should be noted to the experts that the author put the robustness test results and the heterogeneity analysis results in the same table in the previous draft, which may cause the test results to not look very concise and clear. Now, the author has re-stated the robustness test part separately, trying to make the research concise and clear. The following are the regression results of the robustness test:

Table 8 Robustness test results

 (1) (3)

 solow_tfp tfp

b_rd 0.03639** 

 (0.01485) 

b_rd2 -0.0006** 

 (0.00029) 

rd_human 13.74075***

 (4.80635)

rd_human2 -23.83499*

 (12.73865)

ind -1.72343*** 3.78404***

 (0.49572) (1.1974)

urb -0.55251 2.84527

 (0.64188) (1.79191)

mar 0.17864** 0.50752***

 (0.07389) (0.15389)

wage 0.00001*** -8.07e-07

 (1.29e-06) (4.81e-06)

cons 0.33059 -3.20343**

 (0.46437) (1.33754)

Obs 341 341

R2 0.19771 0.10875

Q10:The results should be discussed more in detail.

Modification instructions: The authors are grateful and value the expert opinions. According to the reviewers' opinions, the author discussed, supplemented and enriched the interpretation of the regression results in more detail, and tried to express the research content concisely and comprehensively. Partial empirical analysis results are as follows:

 Table 4 presents the estimated results for the impact of basic research investment on the efficiency of regional economic growth. By gradually introducing control variables to verify the impact of basic research investment on economic growth effects, it is found that the sign and significance of the core explanatory variables and control variables do not change significantly, which indicates that the fixed effect estimation results show strong robustness. From the regression results of models (1) - (5), the coefficients of basic research investment are positive and pass the significance test at the level of 5%, thus indicating that the increase in the proportion of basic research investment contributes to the improvement of regional economic growth efficiency, which is consistent with the research conclusions of Yan and Gong (2013). The coefficient of the core explanatory variable has not changed in models (1) - (5), thus suggesting that the empirical results are robust to some extent. Meanwhile, the regression coefficients of the quadratic terms of the proportion of investment in basic research (b_rd2) are all negative, and they have passed the significance test at the 5% level, which indicates that the proportion of investment in basic research has an "inverted U-shaped" effect on the regional economic growth efficiency path. Thus, Hypothesis I is verified. The possible reasons for this path of influence are that on the one hand, an appropriate increase in the proportion of basic research investment can optimize the efficiency of scientific research resource allocation, and achieve high-efficiency growth of the regional economy by improving the ability of industrial independent innovation. On the other hand, if the proportion of basic research investment is too high, it may lead to an imbalance between basic research and applied research, affect the efficiency of the whole scientific research system from "knowledge technology to productivity", and eventually deteriorate the efficiency of regional economic growth.

Q11:Comparisons with previous studies are absent.

Modification instructions: The author attaches great importance to the suggestions of experts. According to the reviewers' opinions, the author supplemented the comparison and analysis of the existing literature in the empirical results test part.

Q12:Conclusions are too short.

Modification instructions: In the light of expert opinion, the authors have made substantial changes to the "Conclusions part". The result of the modification is as follows:

This study takes the current Chinese R&D resource allocation structure as the research object, and analyses the relationship between the proportion of basic research investment and economic growth efficiency. Simultaneously, it conducts an empirical test on the relationship and its action path with the help of panel data from 31 provinces in the fixed effect model and the mediation effect test model from 2009 to 2019. This study finds that the increase in the proportion of basic research investment can improve the efficiency of regional economic growth, but there is an inverted U-shaped effect path that promotes first and then suppresses, which is supported by a series of subsequent robustness tests. The test results of the threshold model further point out that, with the transition of economic development stages, investment in basic research will play a more significant role in promoting the efficiency of economic growth. The results of the intermediary mechanism test show that an increase in the proportion of investment in basic research can improve the efficiency of regional economic growth by increasing the level of human capital, which shows a partial intermediary effect. The increase in the proportion of investment in basic research has a promoting effect on the increase of regional knowledge reserves, but the path of increasing knowledge reserves has not been supported by empirical data, thus affecting the efficiency of regional economic growth. The regional heterogeneity test results reveal that the research conclusion in the eastern and central regions is consistent with the regression conclusion of the national sample, that is, the increase in the proportion of basic research investment can positively improve the regional economic growth efficiency and show an inverted U-shaped effect path. The regional research conclusions are in complete contrast. The proportion of basic research investment has a negative impact on regional economic growth efficiency, but it does not pass the significance test. Furthermore, under the guidance of the regional non-balanced development strategy, the overall industrial structure adjustment in the western region lags behind, and technological change is slow, thus resulting in a large "technical potential gap" between it and the frontier. This pattern can make better use of the late-mover advantage given the technical potential difference.

Q13:Policy implications are weak.

Modification instructions: The author agrees with the experts. And carefully revised the conclusion section of the study. The results of the revision of the policy implications are as follows:

This study has the following policy implications: Firstly, local governments should be aware of the core driving role of basic research activities for economic growth. They should not only pay attention to the growth of total R&D investment to seek new technological innovation and industrial catch-up but also ensure the optimization of resource allocation of R&D activities in the era of the knowledge economy. It is necessary to tap the innovation potential brought by resource optimization in R&D departments by increasing the proportion of basic research investment. Secondly, in the process of adjusting the resource allocation pattern of R&D activities, on the one hand, we should pay attention to the deployment of scientific research forces in emerging strategic industries. On the other hand, it is necessary to open up the links among enterprises, R&D institutions and universities, build a tripartite benefit sharing mechanism, and improve the efficiency of technological innovation. Thirdly, in the context of accelerating the upgrading of industrial structure and breaking the blockade of technology sources, relevant departments must emphasize the scientific and cultural education in the region and strengthen the cultivation and accumulation of human capital. On this basis, by virtue of the corresponding talent welfare policies, we will attract high-quality labour into the region and support the efficient growth of the regional economy. Fourthly, the eastern coastal areas of China need to speed up the adjustment of the allocation pattern of scientific research resources and form an independent core controllable leading industrial chain and R&D innovation chain by strengthening investment in basic research projects. 

Q14:Further research should be indicated.

Modification instructions: The author agrees with the expert opinion. In the conclusion and policy implications of the study, the author adds the prospect of further research. The modified results are as follows:

This study explores the relationship between the proportion of basic research input and the efficiency of regional economic growth from the provincial macro level. Although there are some innovations in the theoretical analysis framework and research perspective, there are three limitations: (1) The data only use China's panel data from 2009 to 2019, thus lacking data and empirical analysis from other economies. (2) The sample size of provincial level research is small, so future research should use more micro level data to support our conclusions. This study also lacks the test and analysis of the relationship between the two from the perspective of spatial correlation. (3) Although it investigates the impact path of basic research investment on regional economic efficiency from the perspectives of human capital cultivation and basic knowledge enrichment, there may be other impact paths and intermediary variables. Simultaneously, the follow-up research can further build a mathematical analysis model to more comprehensively investigate and analyse the impact of the proportion of basic research investment on regional economic growth efficiency.

Q15:Limitations of the study are not provided.

Modification instructions: The author agrees with the expert opinion. In the conclusion and policy implications of the study, the author adds the limitations and shortcomings. The modified results are as follows:

This study explores the relationship between the proportion of basic research input and the efficiency of regional economic growth from the provincial macro level. Although there are some innovations in the theoretical analysis framework and research perspective, there are three limitations: (1) The data only use China's panel data from 2009 to 2019, thus lacking data and empirical analysis from other economies. (2) The sample size of provincial level research is small, so future research should use more micro level data to support our conclusions. This study also lacks the test and analysis of the relationship between the two from the perspective of spatial correlation. (3) Although it investigates the impact path of basic research investment on regional economic efficiency from the perspectives of human capital cultivation and basic knowledge enrichment, there may be other impact paths and intermediary variables. Simultaneously, the follow-up research can further build a mathematical analysis model to more comprehensively investigate and analyse the impact of the proportion of basic research investment on regional economic growth efficiency.

Q16:Proofreading by a native speaker is required.

Modification instructions: After completing the revision of the sutdy, the author invited the relevant English professional person to check and adjust the expressions.

Q17:The editing does not follow the journal’s guidelines.

Modification instructions: After completing the content modification, the author has made preliminary formatting adjustments to the study in accordance with the format standards of the PLOS ONE.

Q18:The originality value of the study is limited.

Modification instructions: The author believes that the study has the following original values: first of all, although the existing literature has achieved fruitful results in basic research investment to promote economic growth, technological progress and improve regional innovation ability (Sun and Xu, 2017; Li et al., 2018; Magazzino et al., 2021). However, most of these studies start from the perspective of the overall R&D resource investment or the total amount of basic research investment, and do not pay attention to the structural optimization of R&D investment. As R&D activity itself is a complex system that simultaneously includes basic research, applied research, and experimental development research, if the economic effect of R&D activity is investigated only from the total R&D investment itself, it is not only difficult to conduct more in-depth mechanism research and analysis on how R&D activities drive the improvement of economic efficiency, but also easy to lead to the internal resource mismatch of R&D activities. Secondly, China is a typical developing country. In the past four decades of rapid development, China's scientific research resources have been mainly used for applied research, while less resources have been allocated to basic research activities. However, with the total amount and quality of China's economic development approaching the world's cutting-edge level, the adjustment of industrial structure and the improvement of innovation ability will rely more on basic research activities. Therefore, adjusting the allocation structure of scientific research resources is very important and urgent for China's economic development and even the healthy development of the world economy. Under this background, this paper explores the impact of R&D investment structure adjustment on the efficiency of regional economic growth, its channels, and its heterogeneity effect. Finally, the rapid growth of China's economy in the past 40 years is a topic of great interest to the economic community, and the empirical research based on China can enrich the existing endogenous growth theory, innovation driven economy and other related research. In the short term, the increase of the proportion of basic research means the decrease of applied research investment. What is the comprehensive impact on the efficiency of economic growth? Is there an inverted U-shaped relationship? Is the eastern region of China on the left or right side of the turning point of the inverted-U relationship? The answers to these questions will not only help to clarify the mechanism of the impact of basic research investment on economic growth efficiency in theory, but also provide scientific guidance for regions to adjust the structure of R&D investment, optimize the allocation of scientific research resources, and boost high-quality economic development.

Q19:This is a basic econometric exercise without a clear innovative intuition.

How does the paper enrich the knowledge of the scientific community?

Modification instructions: The author think that this study is not a basic econometric exercise without a clear innovative intuition. What the author wants to explain to the experts is: first of all, although the existing literature has achieved fruitful results in basic research investment to promote economic growth, technological progress, and improve regional innovation ability (Sun and Xu, 2017; Li et al., 2018; Magazzino et al., 2021). However, most of studies start from the perspective of the overall R&D resource investment or the total amount of basic research investment, and do not pay attention to the structural optimization of R&D investment. As R&D activity itself is a complex system that simultaneously includes basic research, applied research, and experimental development research, if the economic effect of R&D activity is investigated only from the total R&D investment itself, it is not only difficult to conduct more in-depth mechanism research and analysis on how R&D activities drive the improvement of economic efficiency but also easy to lead to the internal resource mismatch of R&D activities. Secondly, China is a typical developing country. In the past four decades of rapid development, China's scientific research resources have been mainly used for applied research, while less resources have been allocated to basic research activities. However, with the total amount and quality of China's economic development approaching the world's cutting-edge level, the adjustment of industrial structure and the improvement of innovation ability will rely more on basic research activities. Therefore, adjusting the allocation structure of scientific research resources is very important and urgent for China's economic development and even the healthy development of the world economy. Under this background, this paper explores the impact of R&D investment structure adjustment on the efficiency of regional economic growth, its channels, and its heterogeneity effect. Finally, the rapid growth of China's economy in the past 40 years is a topic of great interest to the economic community, and the empirical research based on China can enrich the existing endogenous growth theory, innovation driven economy and other related research.

Reviewer #2

Q1:Abstract: should be written again in academic forms. (Main aim, where study conducted, model used, period, and main findings). Statements are missing.

Modification instructions: The author is very grateful for the valuable comments provided by the experts. According to the expert requirements, the author has made a large range of revisions to the expression of the abstract part. The modification result is as follows:

Abstract: This study explores the relationship between the proportion of basic research investment in R&D expenditure and regional economic growth efficiency and its influence path. A panel of data form 31 Chinese provinces was analysed by the fixed-effects model over 2009 to 2019. Empirical results highlight that: (1) The increase in the proportion of basic research investment has a significant promoting effect on regional economic growth efficiency, but it presents an "inverted U-shaped" impact path. Simultaneously, the threshold effect model test results reveal that basic research investment plays a more significant role in promotion when the per capita income level exceeds a certain threshold. (2) The test results of the mediating effect model indicate that the improvement of the level of human capital is the intermediate channel through which the investment in basic research affects regional economic growth efficiency, while the mediating effect of the increase of knowledge storage on the process of the investment in basic research affecting regional economic growth efficiency is not supported by data. The relevant policy implications are as follows: when the per capita income level of a region rises to a certain stage, the direction of research development should be changed from attaching importance to applied research and foreign technology import to coordinating applied research and basic research, to form independent innovation ability and improve the efficiency of regional economic growth.

Q2:Introduction: The author (s) should highlight the contribution of this work to the existing literature review in the last part of the introduction. The written contribution is not clear and the statements are too long.

Modification instructions: We are very grateful to the anonymous reviewers for their comments. The author has made major changes to the introduction in accordance with the expert suggestions. The revised results of the introduction are as follows:

Since its reform and subsequent opening up, China's economy has developed rapidly and has achieved the "China Miracle", which has attracted global attention. In this process, the "introduction, absorption, and re-digestion" model, represented by imported equipment, foreign direct investment (FDI), and purchase of foreign technology, has played an important role in improving China's technological level and ensuring rapid economic growth. However, as the gap between China's economic development and the world's frontier level continues to narrow, the latecomer advantage that could have been earned from the experience gained and lost through the industrial, technological, and economic advancement of developed countries has gradually disappeared. From the perspective of the external environment, developed economies strictly limit the spill-over of high-technology and advanced knowledge, retain high-end productive service industries in the process of international industrial division, outsource non-core businesses, and firmly grasp the key technologies in the industrial chain and the stagnant core technologies. This situation means that it is increasingly difficult to directly absorb the technology and knowledge of developed economies. From the perspective of the internal environment, China's economy has entered a new stage, and the previous simple technology import mode is beginning to appear "powerless" in the face of pain points in the process of technological development. As the source of all scientific and technological innovation activities, basic research is not only the breakthrough point of various technical problems but also related to China's overall science and technology strategy. The 19th National Congress of the Communist Party of China pointed out that it is necessary to aim for the forefront of global scientific and technological development and to achieve breakthroughs in original achievements by strengthening basic research, to provide strong support for strategies such as building a strong country in science and technology and in quality. Once China's economic development has entered a new stage, whether the traditional scientific research resource allocation pattern of "emphasising application research but ignoring basic research" can adapt to the situation of economic development and improve total factor productivity is a question worthy of careful consideration.

The possible marginal contributions of this study are as follows. First, from the research perspective, unlike the existing literature that focuses on the overall R&D activities or the total amount of basic research investment, this study attempts to explore the factors that may improve the efficiency of regional economic growth from the structural perspective of R&D resource investment. Second, in terms of theoretical analysis, it constructs an analytical framework between the proportion of basic research investment in R&D activities and the efficiency of regional economic growth. Taking China, a typical representative of developing countries, as the research sample, the provincial level data of China from 2009 to 2019 are selected for empirical tests. Finally, in the empirical analysis, this study comprehensively uses the intermediary effect model, threshold effect model, and other empirical methods to carefully test the intermediary channels and various heterogeneous effects on the proportion of basic research investment that affects the efficiency of regional economic growth.

Q3:Literature review: This needs to discuss recent literature to develop the theoretical argument and gaps.

Modification instructions: The author agrees with the expert opinion very much. Therefore, the author adds the literature review section in the study, and discusses the latest relevant literatures to develop the theoretical argument and gaps. For example, the following documents have been added:

1.Mele M, Magazzino C, Schneider N, et al. Innovation, income, and waste disposal operations in Korea: evidence from a spectral granger causality analysis and artificial neural networks experiments[J]. Economia Politica, 2022: 1-33.

2.Magazzino C, Alola A A, Schneider N. The trilemma of innovation, logistics performance, and environmental quality in 25 topmost logistics countries: A quantile regression evidence[J]. Journal of Cleaner Production, 2021, 322: 129050.

3.Ye Jingjing, Zhou Xiaoyao, Chen Shi. Innovative transformation of basic research input -- Based on the evidence supported by the National Natural Science Foundation of China [j] Economics (quarterly), 2021,21 (06): 1883-1902.

4.Zeng Deming, Zhao Shengchao, Ye Jiangfeng, Yang Liang. Basic research cooperation, applied research cooperation and enterprise innovation performance [j] Scientific research, 2021,39 (08): 1485-1497.

In view of the excessive revision of the study, the author suggests experts to check the modification results in Revised Manuscript with Track Changes.

Q4:Research Methods: reason behind the sample selection, period, and models used should be explained in detail. Which panel model been used? Diagnostic tests should be performed to confirm that the model is BLUE (best linear unbiased estimator), clear of heteroscedasticity and autocorrelation.

Modification instructions: The authors very much agree with the experts. Firstly, based on expert opinion, the authors added reasons for the sample selection in the "Data source and variable description" section. The reason for choosing China's 2009-2019 provincial-level panel data is mainly as follows: (1) A moderate cutting-edge technology gap is an important condition for research on the mechanism of basic research investment to promote economic growth. China is a rapidly growing economy around the world over the past 30 years, and the technological level of China's economic growth process is constantly approaching the world's cutting-edge level. Therefore, the selection of China as the research object helps to observe the driving mechanism played by the basic research institute in the process of gradually changing economic and economic growth from the "technology introduction and absorption model" to the "independent research and development innovation model". (2) In 2009, China's per capita national income and per capita GDP exceeded 3,000 dollars, and reached the forefront of middle-income countries according to the standards of the United Nations and the World Bank, thereby indicating that China's industrial structure is constantly optimizing in the process of economic growth. In the process, the impact of basic research on the efficiency of economic growth began to become more and more important. Therefore, 2009 is selected as the starting point for the study sample. (3) Considering the availability of data, especially the relevant data in the "China Science and Technology Statistical Yearbook ", which was only updated to 2019 when the author wrote, this study chose 2019 as the endpoint of the research sample. Secondly, for the choice of model, this study uses the Hausman test, Boostrap Hausman test, Modified Wald test, F test and the economic significance of the research topic to make a comprehensive judgment on the model selection. At the same time, the authors also made relevant supplementary explanations for the model selection part. Based on the Hausman test and the Boostrap Hausman test, the fixed-effects model and the random-effects model are judged, and the fixed-effects model is finally selected based on the test results. Further, an individual fixed-effects model was selected by the F-test results and the economic significance of the model itself. Then, the Modified Wald test is used to determine whether the model has an intergroup heteroscedasticity problem. The test results show that there is heteroscedasticity between groups, so the robustness standard error is used in the model estimation to overcome it.

Table 3 Diagnostic tests results

 Statistics p-value

Hausman test 13.44 0.0195

Boostrap Hausman test 13.76 0.0324

Modified Wald test 25.94 0.7243

Q5:Empirical results and discussion: this section needs better discussion both statistically and theoretically.

Modification instructions: Based on expert opinion, the authors have re-analyzed the empirical analysis results in an effort to provide a more complete theoretical and statistical analysis. Partial empirical analysis results are as follows:

Table 4 presents the estimated results for the impact of basic research investment on the efficiency of regional economic growth. By gradually introducing control variables to verify the impact of basic research investment on economic growth effects, it is found that the sign and significance of the core explanatory variables and control variables do not change significantly, which indicates that the fixed effect estimation results show strong robustness. From the regression results of models (1) - (5), the coefficients of basic research investment are positive and pass the significance test at the level of 5%, thus indicating that the increase in the proportion of basic research investment contributes to the improvement of regional economic growth efficiency, which is consistent with the research conclusions of Yan and Gong (2013). The coefficient of the core explanatory variable has not changed in models (1) - (5), thus suggesting that the empirical results are robust to some extent. Meanwhile, the regression coefficients of the quadratic terms of the proportion of investment in basic research (b_rd2) are all negative, and they have passed the significance test at the 5% level, which indicates that the proportion of investment in basic research has an "inverted U-shaped" effect on the regional economic growth efficiency path. Thus, Hypothesis I is verified. The possible reasons for this path of influence are that on the one hand, an appropriate increase in the proportion of basic research investment can optimize the efficiency of scientific research resource allocation, and achieve high-efficiency growth of the regional economy by improving the ability of industrial independent innovation. On the other hand, if the proportion of basic research investment is too high, it may lead to an imbalance between basic research and applied research, affect the efficiency of the whole scientific research system from "knowledge technology to productivity", and eventually deteriorate the efficiency of regional economic growth.

In terms of control variables, the influence coefficient of industrial structure (ind) on the efficiency of economic growth is positive and has passed the significance test at the 1% level, which indicates that an increase in the proportion of industrial structure has a positive impact on the efficiency of regional economic growth. The possible explanations are as follows: according to Baumol's cost disease theory, the service industry has a lower labour productivity compared with industry. When the regional industrial structure evolves to the tertiary industry, it is easy to cause the economy to fall into a "structural slowdown". The economic growth in most parts of China is still driven by the single engine of industrial development and has not yet switched to the "dual engine driven" growth mode of industry and service industry. The regression coefficient of urbanisation level (urb) is positive and has passed the significance test at the 1% level, which indicates that the improvement in urbanisation level can significantly improve the efficiency of regional economic growth. The reason for this phenomenon is that as China's urbanization process switches from the investment driven and government led mode to a new stage in which the market determines the allocation of resources, a higher level of urbanization provides a greater competitive market and evolutionary incentives for enterprise agglomeration, market competition, and technical cooperation. The influence coefficient of marketization level (mar) on economic growth efficiency is positive and has passed the significance test at the 1% level, which implies that the improvement in marketization level can positively affect regional economic growth efficiency. It may be because the deepening of the marketization process can promote the rationalization of factor allocation, correct the misallocation of resources among industries, and promote structural transformation and upgrading, which will help improve the efficiency of economic growth. The regression coefficient for labour level (wage) is positive but does not pass the significance test.

Table 4 Benchmark regression results

 (1) (2) (3) (4) (5) (6)

 tfp tfp tfp tfp tfp tfp

b_rd 0.1154*** 0.1105** 0.1169** 0.1094** 0.1078** 

 (0.0353) (0.0421) (0.0461) (0.0463) (0.0462) 

b_rd2 -0.0021*** -0.0020** -0.0024** -0.0025** -0.0025** 

 (0.0007) (0.0008) (0.0009) (0.0009) (0.0009) 

ind -0.2820 2.6970** 3.3001*** 3.3866*** 2.5698**

 (0.6601) (1.1215) (1.1431) (1.2139) (1.1893)

urb 5.4002*** 3.0484** 2.7224 6.2194***

 (0.9245) (1.2334) (1.8239) (2.4131)

mar 0.5388*** 0.5299*** 0.5045***

 (0.1593) (0.1592) (0.1914)

wage -1.18e-06 8.52e-06

 (4.86e-06) (5.71e-06)

_cat#c.b_rd 0 0.0149

 (0.0167)

_cat#c.b_rd 1 0.1122***

 (0.0394)

cons 0.9842*** 1.1362** -3.1966*** -2.6583** -2.4287* -4.1030**

 (0.1962) (0.4855) (1.1116) (1.1174) (1.4130) (1.7313)

Obs 341 341 341 341 341 341

R2 0.0285 0.0287 0.0757 0.0997 0.0999 0.1070

Note：*** express p<0.01, * *express p<0.05, * express p<0.1；Standard errors are in parentheses, the same below。

---

## [Decision Letter · Decision Letter 1]

2 Oct 2022

PONE-D-22-13395R1Basic Research Investment, Innovation Capability Improvement and Economic Growth EfficiencyPLOS ONE

Dear Dr. hui,

Thank you for submitting your manuscript to PLOS ONE. After careful consideration, we feel that it has merit but does not fully meet PLOS ONE’s publication criteria as it currently stands. Therefore, we invite you to submit a revised version of the manuscript that addresses the points raised during the review process.

We look forward to receiving your revised manuscript.

Kind regards,

Bing Xue, Ph.D.

Academic Editor

PLOS ONE

Reviewers' comments:

Reviewer's Responses to Questions

**Comments to the Author**

1. If the authors have adequately addressed your comments raised in a previous round of review and you feel that this manuscript is now acceptable for publication, you may indicate that here to bypass the “Comments to the Author” section, enter your conflict of interest statement in the “Confidential to Editor” section, and submit your "Accept" recommendation.

Reviewer #1: (No Response)

2. Is the manuscript technically sound, and do the data support the conclusions?

Reviewer #1: No

3. Has the statistical analysis been performed appropriately and rigorously? 

Reviewer #1: No

4. Have the authors made all data underlying the findings in their manuscript fully available?

Reviewer #1: No

5. Is the manuscript presented in an intelligible fashion and written in standard English?

Reviewer #1: No

6. Review Comments to the Author

Reviewer #1: Literature review must be updated: 10.1016/j.jup.2021.101256; 10.1080/15567249.2020.1868622.

Descriptive statistics are incomplete.

The structure of the manuscript is confusing: Table 2 should be placed before Table 1...

Comparisons with previous results, discussion, Conclusions, policy implications, limitations of the results are too short.

7. PLOS authors have the option to publish the peer review history of their article (what does this mean?). If published, this will include your full peer review and any attached files.

Reviewer #1: No

---

## [Author Response · Author response to Decision Letter 1]

19 Oct 2022

Q1: Literature review must be updated:10.1016/j.jup.2021.101256;10.1080/15567249.2020.1868622.

Modification instructions: The author attaches great importance to expert advice. According to experts' opinions. Authors carefully read the above two papers, deepened their understanding of ICT penetration, power consumption, environmental pollution, urbanization and economic growth, and added the above two papers in the literature review section of the paper.

Q2: Descriptive statistics are incomplete.

Modification instructions: Thanks very much for the opinions of the reviewers. The author adjusted the descriptive statistics of the main variables (Table 1) according to the opinions of the reviewers to make the contents of the table complete.

Q3:The structure of the manuscript is confusing: Table 2 should be placed before Table 1...

Modification instructions: Based on expert opinion, the author rearranged the structure and position of the chart. And all the tables and figures involved in the article are checked, and strive to make the structure of the article clear, not confusing.

Q4: Comparisons with previous results, discussion, Conclusions, policy implications, limitations of the results are too short.

Modification instructions: Many thanks to the experts for their valuable advice. According to the expert opinion, the author added the discussion of the existing literature in the corresponding part of the paper, and made a comprehensive and detailed revision to the conclusion, policy suggestion and article deficiency, so as to make the content more abundant and the logic more smooth. It is hoped that the modification results can meet the requirements of experts.

Q5: Is the manuscript presented in an intelligible fashion and written in standard English?

Modification instructions: According to the expert opinion, the author revises and adjusts the language expression of the full text, and strives to convey the author's views accurately and clearly.

---

## [Decision Letter · Decision Letter 2]

25 Nov 2022

Basic Research Investment, Innovation Capability Improvement and Economic Growth Efficiency

PONE-D-22-13395R2

Dear Dr. hui,

We’re pleased to inform you that your manuscript has been judged scientifically suitable for publication and will be formally accepted for publication once it meets all outstanding technical requirements.

Kind regards,

Bing Xue, Ph.D.

Academic Editor

PLOS ONE

Additional Editor Comments (optional):

Reviewers' comments:

Reviewer's Responses to Questions

**Comments to the Author**

1. If the authors have adequately addressed your comments raised in a previous round of review and you feel that this manuscript is now acceptable for publication, you may indicate that here to bypass the “Comments to the Author” section, enter your conflict of interest statement in the “Confidential to Editor” section, and submit your "Accept" recommendation.

Reviewer #1: (No Response)

2. Is the manuscript technically sound, and do the data support the conclusions?

Reviewer #1: (No Response)

3. Has the statistical analysis been performed appropriately and rigorously? 

Reviewer #1: (No Response)

4. Have the authors made all data underlying the findings in their manuscript fully available?

Reviewer #1: (No Response)

5. Is the manuscript presented in an intelligible fashion and written in standard English?

Reviewer #1: (No Response)

6. Review Comments to the Author

Reviewer #1: ---------------------------------------------------------------------------------------------------------------------------------

7. PLOS authors have the option to publish the peer review history of their article (what does this mean?). If published, this will include your full peer review and any attached files.

Reviewer #1: No

---

## [Editor Report · Acceptance letter]

2 Dec 2022

PONE-D-22-13395R2 

Basic Research Investment, Innovation Capability Improvement, and Economic Growth Efficiency 

Dear Dr. hui:

I'm pleased to inform you that your manuscript has been deemed suitable for publication in PLOS ONE. Congratulations! Your manuscript is now with our production department. 

Kind regards, 

on behalf of

Professor Bing Xue 

Academic Editor

PLOS ONE